# Haemoglobin polymorphism in different blood levels of Ethiopian Zebu x Holstein Friesian crossbred dairy cattle

Eyerusalem Tesfaye[1,2]*, Aberra Melesse[1], Simret Betsha[1], Dereje Andualem[2]

**1** Hawassa University School of Animal and Range sciences, Hawassa, Ethiopia, **2** Dilla University Department of Animal and Range Sciences, Dilla, Ethiopia

* tesfayeeyerusalem@gmail.com

## Abstract

The present study aimed to investigate the biochemical polymorphism of haemoglobin (Hb) among three different blood levels (50%, 75%, and 87.5%) of Ethiopian Zebu x Holstein Friesian (HF) crosses at three milkshed locations by using horizontal agarose gel electrophoresis. To this effect, 117 crossbred lactating cows (39 cows per location namely Shashemene, Hawassa, and Dilla) were used. In each location, thirteen crossbred dairy cows were sampled from each blood level. The red blood cells were separated, washed and lysed following standard procedures. Then, the haemoglobin was typed using gel electrophoresis. The results indicated that the allelic frequencies of 0.89 and 0.11 for $Hb^A$ and $Hb^B$, respectively. The corresponding genotype frequencies were 0.84, 0.11, and 0.05 for $Hb^{AA}$, $Hb^{AB}$, and $Hb^{BB}$ in the tested population, respectively. The Chi-square test revealed that the sampled population was not under Hardy-Weinberg equilibrium. The level of heterozygosity ranged from 0.03 in HF87.5% to 0.18 in HF50% in the studied milkshed dairy cows. The inbreeding coefficient (FIS) calculated for each subpopulation and then pooled across subpopulations, yielded values of 0.38 for blood-level groups and 0.41 for location-based groups, indicating substantial inbreeding within subpopulations. Although the Genetic differentiation (FST) value among subpopulations for blood level was numerically higher (0.06) than FST values for location (0.01), yet both estimates fall within the range typically interpreted as very low differentiation. In conclusion, the various genetic diversity measures and fixation indices revealed the existence of a reduced genetic diversity in the studied population at the biochemical level, which can be serve as foundational information to examine the breeding program for the crossbred dairy cattle in the studied milkshed.

**Data availability statement:** Yes - all data are fully available without restriction; Relevant data are within the paper and its Supporting information files. The minimal dataset has also been uploaded to https://doi.org/10.5281/zenodo.18782494.

**Funding:** The author(s) received no specific funding for this work.

**Competing interests:** The authors have declared that no competing interests exist.

## Introduction

According to the Central Statistical Agency [1] the Ethiopian agricultural community keep cattle primarily for milk, draught power, breeding, and beef production. The same source states that livestock producers hold a larger population of local breeds, which account for 96.8% of all cattle in the nation, and the remaining are the cross-breds and exotic ones accounted for about 2.71 and 0.41 percent, respectively. About 12.8 million milking cows consisting of indigenous, exotic and their crosses are used for dairying purpose [1].

Despite the technical, institutional, and socio-economic factors affecting its productivity, the dairy sector still significantly contributes to the national economy because of climate favorable locations with lower animal disease stress [2]. Despite their extremely low productivity per animal as compared to the exotic and their crosses, that the indigenous dairy cattle breeds are also substantially contributing to the national dairy development of Ethiopia [3]. To benefit from the genetic variations of the combination effect of exotic and indigenous cattle breeds through crossbreeding, the country has been importing different exotic cattle breeds of which the main being the Holstein-Friesian cattle breed [4,5].

Genetic variation has been studied via blood protein analysis as one of the biochemical approaches to investigate the polymorphism occurring at the protein level [6,7]. Due to the significance of biochemical polymorphic character studies in the improvement of farm animals as well as the possibility that certain polymorphic alleles tend to be linked to economically important traits through general heterozygosity or the pleiotropic effect, studies focusing on these polymorphic traits have become vital [8]. Haemoglobin is a blood protein that has been defined as an essential iron-containing protein responsible for oxygen transport known to exhibit polymorphism at its globin portion in various livestock species [9].

The detection of biochemical polymorphisms in gene products at structural loci provides a precise procedure to localize and prove their reliability as genetic markers for some economic traits and livestock diseases [10]. Haemoglobin polymorphism is one of the biochemical markers, which is a gene-controlled diversity in the different farm animals due to variation in the amino acid sequence in the polypeptide chains of haemoglobin [8]. Accordingly, haemoglobin has been proven to have biochemical, biophysical, and physiological properties, and its polymorphic variants have been reported to be associated with morphological, performance, and adaptation traits in cattle [11,12]. Despite the feasibility of studying biochemical protein variants such as haemoglobin in Ethiopia, investigations into haemoglobin polymorphism remain scarce. The few available studies are breed-specific, including research on Ogaden cattle [11] and earlier work by [13]. For example, limited studies have revealed the existence of three haemoglobin genotypes namely $Hb^{AA}$, $Hb^{AB}$, and $Hb^{BB}$ in Ogaden cattle of Southeastern Ethiopia and Bunja cattle of Nigeria [14].

According to [15,16], the Shashemene-Dilla milkshed is a highly contributing region with a significant number of crossbred dairy cattle with unknown genetic composition. Consequently, keeping a dairy herd with unknown genetic makeup might lead to the deterioration of the future performance of the diary sector due

accumulation of inbreeding and other negative effects. This calls for a study on the genetic diversity of the dairy cattle population in the region. Although DNA-based technologies have become increasingly accessible, protein polymorphism methods remain practical in settings with limited molecular laboratory infrastructure because they require simpler equipment and lower reagent costs, while still providing useful genetic information [10,17]. Given that genetic research in many African nations like Ethiopia is not advanced, the significance of this alternative approach in studying animal genetic diversity would be highly beneficial [18]. Therefore, this study aimed at investigating the biochemical polymorphism at haemoglobin locus of different blood levels of Holstein-Zebu crosses reared in Shashemene-Dilla milkshed dairy farms. It has been examined whether the observed haemoglobin genotype frequencies conformed to or deviated from Hardy–Weinberg equilibrium expectations as commonly assessed in population genetic studies.

## Materials and methods

### Ethical approval

Permission was obtained from Hawassa University Research Ethics Review Committee to carry out the present work (HURERC-REC016/23). Two senior veterinary doctors with great experience in standard protocol for animal care and welfare were engaged during sample collection.

### Description of the study area

This research was carried out in milkshed dairy farms located in the area extending from Shashemene to Dilla, which is one of the high milk-producing areas in the Rift Valley of Ethiopia. It is located between 250 and 375 kilometers south of Addis Ababa, on the Addis Ababa–Moyale route. Shashemene-Dilla milkshed is chosen for the study based on the existence of dairy farms maintaining crossbred dairy cattle breeds; each of these locations has its distinct agricultural and social customs.

The major locations considered in the study are Shashemene, Hawassa, and Dilla. Shashemene is located in Oromia Regional State, West Arsi Zone, 250 km south of the capital Addis Ababa. (https://en.m.wikipedia.org/wiki/shashemene). The town lies within at an altitude ranging from 1,794−2,094 meters above sea level. It receives an annual rainfall of 879 mm. The average annual temperature is 26°C. The warmest month of the year is March, with an average temperature of 28°C. In Shashemene, July is the coldest month with an average temperature of 23°C and an annual temperature range of 12–27°C. (https://www.accuweather.com/en/et/shashemene).

Hawassa, the regional capital of Sidama located 275 km south of the capital Addis Ababa along the Addis Ababa–Moyale highway (https://en.m.wikipedia.org/wiki/hawassa). It has an altitude of 1708 masl and is located at 6° 8’ to 7° 17’ N and 38° 24’ to 38° 72’ E. (https://www.accuweather.com/en/et/hawassa).

Dilla is another area considered in this study. It is located 90 km south of the Sidama regional town Hawassa. It is located at 6° 22’ to 6° 42’ N and 38° 21’ to 38° 41’ E and at an altitude range of 1,594 masl (https://en.m.wikipedia.org/wiki/dilla). It receives an annual rainfall of 850 mm and the annual average minimum and maximum temperatures of 12.5°C and 28.0°C, respectively (https://www.accuweather.com/en/et/dilla).

### Blood sample collection and analysis

Blood samples were taken from three distinct blood levels (i.e., 50%, 75%, and 87.5%) of Ethiopian Zebu x Holstein Friesian (HF) crosses that were kept in intensive dairy farms located at Shashemene-Dilla milkshed and whose lineage could be traced back. The total sampled population was 117 from which 39 were drawn from each location namely Shashemene, Hawassa, and Dilla. The sample size of crossbred cows from each blood level in the three locations of the study milkshed was 13. Although 13 cows were sampled per blood level per location, genotype counts in the results reflect the naturally occurring distribution of Hb types within these animals (S1 Data).

Blood was collected using a 5 ml EDTA-coated tube during the morning period by puncturing the jugular vein. Blood collection was done in the morning to have enough time to centrifuge the whole blood sample and wash the RBCs for further haemolysate preparation. During sample collection, two senior veterinarians and 6 assistants were involved in restraining the animal during sampling. Blood samples were immediately put into an ice-box and transported to the Hawassa University molecular biotechnology laboratory and stored at −20 °C until used for the analysis.

Blood analysis and haemolysate preparation was carried out according to the method described by [11,19] with slight modification by keeping the blood samples collected via veno-puncture in a 5 ml vacutainer tube coated with anti-coagulant EDTA.

**Haemolysate preparation.** The analysis took place at Hawassa University's Biotechnology Laboratory in the School of Animal and Range Sciences. The whole blood sample was centrifuged immediately after collection at 3000 rpm/4c° for 5 minutes. The upper liquid portion (supernatant plasma and buffy coat) was removed from the centrifuged EDTA blood and cells were washed with saline solution three times. Saline solution (9.0g of NaCl in 1-liter deionized water) was discarded from the last centrifuge. An equal volume of distilled water and 1/2 volume of carbon tetrachloride ($CCl_4$) was added to the RBC sample and then shaken vigorously. The procedures involving $CCl_4$ were performed under laboratory chemical safety guidelines wearing full personal protective equipment (mask, gloves, and lab coat). The cells with distilled water and carbon tetrachloride were centrifuged for 5 minutes at 1500 G. After the centrifugation, three layers (upper haemolysate, middle RBC leftover and carbon tetrachloride at the bottom) were formed. The upper portion (haemolysate) was used for the determination of agarose gel electrophoretic haemoglobin polymorphism while the remaining was discarded.

**Haemoglobin electrophoretic analysis.** To examine the inherent biochemical variations in haemoglobin (Hb), gel electrophoresis was performed with 1% agarose gel and separated using Tris-EDTA-Borate at pH 8.6. Then, the electrophoresis was carried out for 2 hours at 120 volts [11].

**Identification of haemoglobin polymorphisms.** Based on how quickly the haemoglobin (Hb) bands moved towards the cathode, different Hb phenotypes were distinguished [20]. After the electrophoretic run was completed, the hemoglobin bands were visualized using direct observation and a gel documentation system (Alpha imager Mini Gel Documentation System, For Colorimetric, UV Fluorescent) without discoloration. The Hb polymorphism was detected based on differential speed of migration. A single fast-moving band is designated as a "BB homozygote" (HbBB), whereas a single slow-moving band is indicated by the term "AA homozygote" (HbAA). The existence of both slow- and fast-moving bands was denoted as AB heterozygote (HbAB).

## Statistical analysis

The allele and genotype frequencies (observed and expected) were calculated using GeneAlEx6.503. It was performed from the data obtained in the hemoglobin electrophoretic determination analysis. The observed Hb allele and genotype frequencies were subjected to PopGene 1.32 [21] software to execute a chi-square test for goodness of fit for the observed and expected frequencies under Hardy-Weinberg equilibrium [22]. For the subgroup where the expected number of rare homozygotes (BB) fell below 1 the assumptions of the classical three-class $\chi^2$ test were not met. Therefore, the default collapsed 1-df test was used. The $\chi^2$ and p-values presented in Tables 1 and 2 reflect this adjustment.

Heterozygosity (HE), Homozygosity (HO), effective number of alleles (ne), and Summary of F-statistics (Fixation indices) were also computed using the same software to detect the existence of genetic variability in the crossbred dairy cattle populations. Estimates of heterozygosity between different blood levels of the Holstein crosses were measured as the unbiased estimate of mean heterozygosity [23].

## Results

### Allele and genotype frequencies

The allele and genotype frequencies in three different blood level HF crosses are presented in Table 1. The three distinct hemoglobin genotypes of AA (slow-moving band), AB (both slow and fast-moving band), and BB (fast-moving band) were

**Table 1. Allele and genotype frequencies in three different blood levels of the Ethiopian Zebu x HF crosses.**

| Blood level | Alleles | Allele freq. | Genotypes | Genotype freq. | Genotypes observed | expected | $X^2$, df=1 (*p*=value) |
|---|---|---|---|---|---|---|---|
| HF50% | A<br>B | 0.81<br>0.19 | Hb AA | 0.72 | 28 | 25.4 | |
| | | | Hb AB | 0.18 | 7 | 12.3 | 7.64 (0.01) |
| | | | Hb BB | 0.10 | 4 | 1.36 | |
| HF75% | A<br>B | 0.89<br>0.12 | Hb AA | 0.80 | 32 | 30.5 | |
| | | | Hb AB | 0.13 | 5 | 8.07 | 6.27 (0.01) |
| | | | Hb BB | 0.05 | 2 | 0.47 | |
| HF87.5% | A<br>B | 0.99<br>0.01 | Hb AA | 0.97 | 38 | 38.0 | |
| | | | Hb AB | 0.03 | 1 | 1 | 0.00 (1.0 ns) |
| | | | Hb BB | 0 | 0 | 0 | |
| Entire population | A<br>B | 0.89<br>0.11 | Hb AA | 0.84 | 98 | 93.3 | |
| | | | Hb AB | 0.11 | 13 | 22.4 | 21.45 (<0.001) |
| | | | Hb BB | 0.05 | 6 | 1.29 | |

ns= non-significant (p>0.05).

HF50%=Holstein Friesian with 50% local cattle blood level; HF75%=Holstein Friesian with 25% local cattle blood level; HF87.5%=Holstein Friesian with 12.5% local cattle blood level.

**Table 2. Hb allele and genotype frequencies of different blood levels of Ethiopian Zebu x HF crosses at three locations.**

| Milkshed locations | Alleles | Allele freq. | Genotypes | Genotype freq. | Genotypes observed | expected | $X^2$, df=1 (*p*=value) |
|---|---|---|---|---|---|---|---|
| SHA | A<br>B | 0.94<br>0.06 | Hb AA | 0.9 | 35 | 34.13 | |
| | | | Hb AB | 0.08 | 3 | 4.74 | 6.49 (0.01) |
| | | | Hb BB | 0.03 | 1 | 0.13 | |
| HAW | A<br>B | 0.9<br>0.1 | Hb AA | 0.82 | 32 | 31.36 | |
| | | | Hb AB | 0.13 | 6 | 7.27 | 1.35 (0.25 ns) |
| | | | Hb BB | 0.05 | 1 | 0.36 | |
| DIL | A<br>B | 0.85<br>0.15 | Hb AA | 0.8 | 31 | 27.86 | |
| | | | Hb AB | 0.1 | 4 | 10.29 | 15.72 (<0.001) |
| | | | Hb BB | 0.1 | 4 | 0.86 | |

ns= non-significant (p>0.05).

SHA=Shashemene; HAW= Hawassa; DIL=Dilla milkshed locations.

observed in agarose gel electrophoresis. Tables 1 and 2 represents the genotype and allele frequencies at the haemoglobin locus, as well as the observed and expected numbers of hemoglobin genotypes along with the Chi-square test for the HF crossbred population divided into three groups according to different blood levels and locations. The HbA was the most predominant allele in the studied population (Table 1).

Higher gene and genotypic frequency variations were found among the population categorized in the blood level group rather than in three different locations of the studied milkshed (Table 1). The Chi-square test revealed that except for 87.5% exotic blood levels, the sampled subpopulations were not under Hardy-Weinberg equilibrium for haemoglobin locus (Table 1). The HF crosses at Hawassa (HAW) dairy farms on the other hand were under Hardy-Weinberg equilibrium from the population grouped into locations (Table 2). Haemoglobin was found to be polymorphic across location and level of exotic gene inheritance of cows. The blood level and location also had a significant association (P<0.05) with the occurrence of Hb types except for 87.5% exotic blood level and HF crosses at HAW location.

## The genetic diversity measurements

The percentage of polymorphic loci and measurements of genetic variation at haemoglobin locus between dairy cows of various levels of exotic gene inheritance are shown in Table 3. Both the observed (HO) and expected heterozygosity (HE) at the haemoglobin locus showed that 50% crosses had higher HO and HE while the HO, the HE as well as the homozygosity estimates were similar for 87.5% HF crosses (Table 3).

The observed and expected homozygosity estimates on the other hand showed that the HF crosses with 87.5% and 50% exotic genes had the highest and lowest homozygosity estimates, respectively. The 50% HF crosses had a better effective number of alleles per locus. As shown in Table 4, the highest observed homozygosity was noted in SHA and DIL milk shades with comparable values. However, the lowest expected homozygosity estimate was found in DIL while the highest was in the SHA location. The HF crosses at the Dilla location had a relatively higher effective number of alleles per locus than the SHA and HAW locations.

A measure of Nei's genetic closeness and distance between different blood levels HF crosses is presented in Table 5. The closest genetic relationship was observed between 50% and 75% exotic gene inheritance (D = 0.01), while the farthest genetic distance was found between 50% and 87.5% crosses (D = 0.03).

The highest genetic identity of HF crosses reared in SHA was observed with those of HAW(Table 6). Similarly, the highest genetic similarity was observed between HF crosses reared in HAW and DIL milkshed locations. On the other hand, the Nei's measurement for genetic divergence between HF crossbred populations at three study locations revealed no genetic distance between populations reared in different locations.

**Table 3. Genetic variation measurements at haemoglobin locus in three different blood levels of Ethiopian Zebu x HF crosses.**

| Parameters | Blood levels | | | Pooled |
|---|---|---|---|---|
| | HF50% | HF75% | HF87.5% | |
| Observed heterozygosity | 0.18 | 0.13 | 0.03 | 0.11 |
| Expected heterozygosity | 0.32 | 0.21 | 0.03 | 0.19 |
| Observed homozygosity | 0.82 | 0.87 | 0.97 | 0.89 |
| Expected homozygosity | 0.69 | 0.79 | 0.97 | 0.81 |
| Effective number of alleles per locus (Ne) | 1.45 | 1.26 | 1.03 | 1.24 |
| The percentage of polymorphic locus | 100 | 100 | 100 | 100 |

HF50% = Holstein Friesian with 50% local cattle blood level; HF75% = Holstein Friesian with 25% local cattle blood level; HF87.5% = Holstein Friesian with 12.5% local cattle blood level.

**Table 4. Measurements of genetic variations at haemoglobin locus in different blood levels Ethiopian Zebu x HF crosses reared in different locations.**

| Parameters | Milkshed locations | | |
|---|---|---|---|
| | SHA | HAW | DIL |
| Observed heterozygosity | 0.08 | 0.15 | 0.1 |
| Expected heterozygosity | 0.12 | 0.19 | 0.26 |
| Observed homozygosity | 0.92 | 0.85 | 0.9 |
| Expected homozygosity | 0.88 | 0.81 | 0.74 |
| Effective number of alleles per locus (Ne) | 1.14 | 1.23 | 1.35 |
| The percentage of polymorphic locus | 100 | 100 | 100 |

SHA=Shashemene; HAW= Hawassa; DIL=Dilla milkshed locations.

**Table 5. Measures of Nei's genetic identity (above diagonal) and genetic distance (below diagonal) between different blood levels of Ethiopian Zebu x HF crosses.**

| Blood levels | HF50% | HF75% | HF87.5% |
|---|---|---|---|
| HF50% | *** | 1.0 | 0.98 |
| HF75% | 0.01 | *** | 0.99 |
| HF87.5% | 0.03 | 0.01 | *** |

HF50% = Holstein Friesian with 50% local Zebu cattle blood level; HF75% = Holstein Friesian with 25% local Zebu cattle blood level; HF87.5% = Holstein Friesian with 12.5% local Zebu cattle blood level.

**Table 6. Measures of Nei's genetic identity (above diagonal) and genetic distance (below diagonal) between different blood levels of Ethiopian Zebu x HF crosses reared in different locations.**

| Locations | SHA | HAW | DIL |
|---|---|---|---|
| SHA | *** | 1.0 | 0.99 |
| HAW | 0.00 | *** | 1.0 |
| DIL | 0.01 | 0.00 | *** |

SHA=Shashemene; HAW= Hawassa; DIL=Dilla milkshed locations.

## The fixation indices

The summary of F-statistics (Fixation Indices) results revealed that the inbreeding coefficient value within individuals relative to their subpopulations (FIS) grouped in the three different exotic blood levels and three locations were 0.38 and 0.41, respectively (Table 7). Genetic differentiation among subpopulations grouped by location was very low (FST = 0.01), indicating near-panmictic conditions. Similarly, subpopulations grouped by exotic gene inheritance level also showed low differentiation (FST = 0.06). The inbreeding coefficient value of individual crossbred dairy cow relative to the total population (FIT) was 0.42.

## Discussion

The electrophoresis result of the current study showed the existence of three types of haemoglobin phenotypes namely, a slow-moving (AA), a fast-moving (BB), and a combination of both slow + fast-moving bands (AB). The complex molecules that make up hemoglobin, the oxygen-carrying component of blood, are composed of a protein part, the globin, and an effective haeme part. While the haeme part of the haemoglobin molecule is relatively constant, the globin part, a combination of two sets of polypeptide chains that makes up to 96% of the haemoglobin molecule, varies considerably from species to species and within a species [24].

**Table 7. Summary of F-Statistics (The Fixation indices).**

| Population subgroup | Locus | Sample size | FIS | FIT | FST |
|---|---|---|---|---|---|
| Blood levels | Hb | 234 | 0.38 | 0.42 | 0.06 |
| Locations | Hb | 234 | 0.41 | 0.42 | 0.01 |

Blood level (50%HF, 75%HF, and 87.5%HF); Locations (SHA, HAW, and DIL).

Since the haemoglobin polymorphism is mostly due to the β-chain variant, the - βB shows a Mendelian mode of inheritance presenting 3 phenotypes Hb<sup>A</sup>, Hb<sup>B</sup>, and Hb<sup>AB</sup>, though the rare α-chain variants are also reported in Podolian cattle [25]. The first to describe two electrophoretically different components of haemoglobin in cattle were Cabannes & Serain1 which were reported in Algerian cattle [11]. The Hb<sup>C</sup>, a third component, and Hb<sup>D</sup>, a fourth component were discovered following the start of the starch gel electrophoresis. The Hb<sup>C</sup> variant has been regarded as a typical "African" haemoglobin, because of its relatively high frequency in the breeds and types of African cattle [26].

The Italian Podolic cattle, which is native to Southern Europe, were reported to have two new alpha and one-beta globin variants around the end of the 20th century [27]. Researchers of haemoglobin polymorphism in Grey Alpine cattle formulated the hypothesis of the hemoglobin system as a signature of a past common origin between Grey Alpine and Italian Podolic cattle after they found out the existence of the Y alpha-globin variant (HbAY) and the Zebu beta-globin variant in Grey Alpine cattle [28].

In the present study, the frequency of haemoglobin variant A was highest in cows with higher level of exotic inheritance(87.5%) while the highest frequency Hb<sup>B</sup> was observed in the HF crosses of 50%.. This finding is in agreement with those reported for crosses of HF cattle with different local cattle breeds [29–32]. In earlier studies, [33] reported a significantly higher frequency of Hb<sup>B</sup> in *Bos indicus* type cattle compared to *Bos taurus* type breeds. The Hb<sup>B</sup> exhibited a notably elevated frequency in Zebu cattle, demonstrating a decline in frequency of 3/8–7/8 in European-Zebu hybrids [34]. The gene and genotypic frequency of the current study is consistent with earlier study results, which revealed a high prevalence of Hb<sup>A</sup> in *Bos taurus* cattle [35,36].

The current study's findings regarding the predominance of Hb<sup>AA</sup> type is consistent with those reported for Friesian x Bunaji cattle crosses [30] and Ogaden cattle [11]. As the altitude of the terrain increases in the current milkshed locations, the frequency of Hb<sup>A</sup> tends to increase which is consistent with the reports of [37]. Research by [38] reported that the Hb<sup>A</sup> variant is more prevalent in areas with stressful production environments such as frequent drought and extreme temperatures. In another study, [24] reported the predominance of Hb<sup>A</sup> variant in cows and buffaloes with allelic frequencies of 0.51.

The haemoglobin locus genotypic frequencies, in the tested population of HF crosses in the studied milkshed were not under Hardy-Weinberg equilibrium. The methods of mating, evolutionary influences over time, sample size, migration and artificial selection could be some of the reasons behind this observation [39,40]).

The present study revealed pooled heterozygosity of 0.11, which indicates the presence of a relatively low genetic variation at the hemoglobin locus in the studied crossbred dairy cattle. Low heterozygosity (0.11) is consistent with many studies of blood-protein polymorphisms such as reviews of haemoglobin variation in livestock which reported relatively moderate heterozygosity compared to highly polymorphic DNA markers [8]. Nonetheless, there was variation in the heterozygosity values among the three genotypes and locations.

Alleles of the Hb locus were polymorphic in the studied HF crossbred dairy cattle population and locations. From the F-statistics results, values of the inbreeding coefficient (FIS) within populations grouped in the three different exotic blood levels and three locations were 0.38 and 0.41, respectively. Genetic differentiation among populations was very low for both location (FST = 0.01) and exotic blood level (FST = 0.06), indicating minimal structure. The inbreeding coefficient value of individuals relative to the total population (FIT) was 0.42. The observed genetic diversity among different inheritance of exotic genes (HF 50%, HF 75%, and HF 87.5%) reared in the three different milkshed locations might be associated with their genetic adaptation to the production environment in which they have been managed. Expanded studies linking Hb genotypes with health, milk production, and heat stress traits are encouraged to inform targeted herd management and genetic improvement strategies.

## Supporting information

**S1 Data. HF crosses Haemoglobin data.**

(XLSX)

## Author contributions

**Conceptualization:** Eyerusalem Tesfaye, Aberra Melesse, Simret Betsha.

**Data curation:** Eyerusalem Tesfaye.

**Formal analysis:** Eyerusalem Tesfaye.

**Investigation:** Eyerusalem Tesfaye.

**Methodology:** Eyerusalem Tesfaye.

**Software:** Eyerusalem Tesfaye.

**Supervision:** Aberra Melesse, Simret Betsha, Dereje Andualem.

**Validation:** Aberra Melesse, Simret Betsha, Dereje Andualem.

**Visualization:** Aberra Melesse, Simret Betsha, Dereje Andualem.

**Writing – original draft:** Eyerusalem Tesfaye.

**Writing – review & editing:** Aberra Melesse, Simret Betsha, Dereje Andualem.

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
