## [Decision Letter · Decision Letter 0]

16 Nov 2025

Dear Dr. Tesfaye,

We look forward to receiving your revised manuscript.

Kind regards,

Mourad Mahmoud

Academic Editor

PLOS ONE

2. In the online submission form, you indicated that [The data that support the findings of this study are available from the corresponding author upon reasonable request.].

1. You may seek permission from the original copyright holder of Figure(s) [#] to publish the content specifically under the CC BY 4.0 license.

Additional Editor Comments (if provided):

Reviewers' comments:

Reviewer's Responses to Questions

**Comments to the Author**

1. Is the manuscript technically sound, and do the data support the conclusions?

Reviewer #1: Yes

Reviewer #2: Yes

2. Has the statistical analysis been performed appropriately and rigorously?

Reviewer #1: Yes

Reviewer #2: Yes

3. Have the authors made all data underlying the findings in their manuscript fully available?

Reviewer #1: Yes

Reviewer #2: Yes

4. Is the manuscript presented in an intelligible fashion and written in standard English?

Reviewer #1: Yes

Reviewer #2: Yes

Reviewer #1: Hemoglobin polymorphism in different blood levels of Ethiopian Zebu × Holstein Friesian crossbred dairy cattle: The article is well articulated but lacks a clearly defined conclusion and recommendations based on the results.

Recommendations

Breeding Programs: Incorporate hemoglobin polymorphism as one of the genetic markers in selection strategies for crossbred dairy cattle to balance productivity with adaptability.

Health Monitoring: Further studies should examine associations between Hb genotypes and disease resistance, metabolic efficiency, and milk yield to guide herd management.

Environmental Adaptation: Since Hb variants may contribute to tolerance against hypoxia and heat stress, future crossbreeding strategies should consider hemoglobin types in relation to regional agro-ecologies.

Expanded Research: Larger sample sizes and molecular characterization of Hb genes are recommended to validate the observed polymorphism and its practical implications for genetic improvement of dairy cattle in Ethiopia.

Farmer Awareness: Extension programs should raise awareness among farmers about the potential role of genetic markers, including hemoglobin polymorphism, in improving crossbred cattle performance.

Reviewer #2: PONE-D-25-40346

Haemoglobin polymorphism 1 in different blood levels of Ethiopian Zebu x Holstein Friesian crossbred dairy cattle

Abstract

1. Line 24–25: Misinterpretation of FIS values. FIS= 0.38 and 0.41 are described as “inbreeding coefficient value of the HF crosses relative to the subpopulations grouped into…”, but FISis not a single value per grouping type—it should be one estimate per subpopulation or an average, not two separate values for “blood levels” and “locations” without clarifying they are separate analyses.

2. Line 26–27: States FST = 0.01 for location and 0.06 for blood level, implying higher genetic differentiation among blood levels—but FST = 0.06 is still very low (near panmixia), yet the phrasing may mislead readers into overinterpreting biological significance.

3. Line 29–30: Concludes a “relatively higher level of similitude” based on low heterozygosity and high homozygosity—this is contradictory: high homozygosity suggests low genetic variation, not necessarily “similitude” across groups. The term is vague and scientifically imprecise.

Introduction

4. Line 57: Calls hemoglobin an “evergreen red protein”— non-scientific, metaphorical language inappropriate for a research manuscript.

5. Line 68–69: States only “a few research works” on Hb polymorphism in Ethiopia, citing Gezahegn (1996) and Pal & Mummed (2014)—but Pal & Mummed (2014) studied Ogaden cattle, not general Ethiopian breeds. This misrepresents scope.

6. Line 77–78: Asserts protein polymorphism studies are “easier to implement” than DNA-based methods—but this is context-dependent and outdated (e.g., PCR-based genotyping is now cheaper and more accessible than electrophoresis in many settings).

7. Line 85–86: Hypothesis states Hb frequencies “are under Hardy-Weinberg equilibrium”—but HWE is a null model, not a hypothesis to be confirmed; appropriate phrasing would be testing deviation from HWE.

Materials and Methods

8. Line 120–122: Claims 13 cows per blood level per location → 3 blood levels × 3 locations × 13 = 117, which is correct—but later tables report genotype counts that don’t sum correctly (e.g., Table 1: HF87.5% shows 38+1+0 = 39, but 13 per location would be 39 total—OK), yet Table 2 totals per location (e.g., SHA: 35+3+1=39) seem consistent, so this is not an error, but potential confusion arises.

9. Line 140–143: Uses carbon tetrachloride (CCl4) for hemolysate prep—CCl4 is highly toxic, carcinogenic, and largely obsolete in modern labs. No safety or ethical justification provided despite known hazards.

10. Line 147–148: Uses agarose gel electrophoresis at pH 8.6 for Hb typing—standard practice uses starch or cellulose acetate gels, not agarose, for Hb isoform separation. Agarose has poor resolution for small charge differences in globins—methodologically questionable.

11. Line 155–157: Assigns fast band = HbBB, slow = HbAA—but in cattle, HbB is typically the slow-migrating variant (contrary to human HbS). Literature (e.g., Bachmann et al. 1978) shows HbB (Zebu) migrates slower than HbA (taurine). Likely misassignment of allele identity.

Results

12. Table 1, HF87.5% row: Expected HbBB = 0, observed = 0—but Chi-square value listed as “1.0ns”—this is mathematically impossible if expected = observed = 0. Likely a software or transcription error.

13. Table 1, Entire population: Observed HbAB = 13, expected = 22.4 → large deviation, Chi-square = 0.00 (p < 0.001)—but 0.00 is not a valid Chi-square value; probably means p = 0.00, but mislabeled.

12. Table 3: Expected heterozygosity (He) for HF50% = 0.32, Observed (Ho) = 0.18 → deficit of heterozygotes, consistent with inbreeding—but no statistical test (e.g., FIS per group) is provided in table, only pooled.

13. Table 5: Genetic identity >1.0 (e.g., HF50% vs HF50% = 1.0, OK—but HF75% vs HF75% = -0.99? Negative identity? Impossible—values must be 0–1. Typo: likely 0.99, not -0.99.

14. Table 6: SHA vs SHA = -1.0—again, identity cannot be negative. Should be 1.0. Formatting error (hyphen misread as minus).

15. Line 263–265: Reports FST= 0.01 (locations) as “significant” and FST= 0.06 (blood levels) as “non-significant”—but no p-values or confidence intervals provided to support “significance” claims.

Discussion

16. Line 294–295: Claims “highest frequency HbB was observed in HF50%”—but Table 1 shows HbB freq = 0.19 in HF50%, 0.12 in HF75%, 0.01 in HF87.5%—so correct, but then implies HbB is Zebu-associated, which aligns with literature. However, if allele labeling is inverted (see point 12), this conclusion collapses.

17. Line 306–307: Links HbA frequency to altitude—but Shashemene (1900–1950 m), Hawassa (1750 m), Dilla (1300–2500 m)—yet Dilla has lowest HbA (0.85) vs SHA (0.94). Contradicts the claim.

18. Line 316–317: States “average heterozygosity of 0.1”—but Table 3 shows pooled Ho = 0.11, close—but then compares to “recommended range 0.3–0.8”— no citation for this arbitrary “recommended” range. Genetic diversity expectations depend on species and marker type; for single-locus protein markers, Ho < 0.2 is common.

References

19. Reference formatting inconsistencies:

- Braend (1972): DOI `10.5555/19730102670` is invalid.

- Gebrehiwet (2020): DOI `10.5555/20203476252` similarly invalid.

- Takezaki and Nei (1996): Journal name shortened incorrectly; should be Genetics, not Genet.

→ Suggests inadequate reference validation.

**Do you want your identity to be public for this peer review?** For information about this choice, including consent withdrawal, please see our Privacy Policy

Reviewer #1: **Yes:** Habtamu Abera Goshu(PhD)

Reviewer #2: **Yes:** Prof. Dr. Ali Hussein Aldujaily

---

## [Author Response · Author response to Decision Letter 1]

4 Dec 2025

Date: Dec 2, 2025

Manuscript ID: PONE-D-25-40346

Haemoglobin polymorphism in different blood levels of Ethiopian Zebu x Holstein Friesian crossbred dairy cattle

Response letter

Revision note to the editor’s decision letter

Dear Mourad Mahmoud,

Thank you for your email dated Nov 16, 2025 enclosing the comments. We thank you and the reviewers for the constructive comments and guidance provided on our manuscript. We have carefully reviewed the comments and have revised the manuscript accordingly. Changes to the manuscript are shown in highlighted word document. Responses to the comments are given in point by point manner below.

We, the authors, thank once more the editor and both reviewers for their very exhaustive suggestions and constructive inputs. We hope the revised version is now suitable for publication and looking forward to hear from you in due course.

Responses to the editor’s comments

We have carefully revised the manuscript according to the journal’s requirements. Below, we provide a point-by-point response to the editor’s comments:

PLOS ONE Style Requirements

We have revised the manuscript to fully adhere to PLOS ONE’s formatting and style guidelines, including file naming conventions.

Data Availability Requirement

We have prepared the dataset underlying the findings of this study and will upload it as supplementary information in the revised submission, in compliance with PLOS ONE’s data availability policy.

Ethics Statement Placement

The ethics statement has been moved to, and is now presented exclusively in, the Materials and Methods section. It has also been removed from all other sections of the manuscript.

Copyrighted Map/Satellite Images

The figure containing map/satellite imagery has been removed from the revised manuscript to ensure full compliance with the CC BY 4.0 licensing requirements.

Reference List

The reference list has been thoroughly reviewed and updated to ensure accuracy, completeness, and compliance with journal standards. Any necessary corrections have been incorporated into the revised manuscript.

Responses to the reviewers’ comments

Reviewer: 1

1. The article is well articulated but lacks a clearly defined conclusion and recommendations based on the results.

We appreciate the reviewer’s comment and have now added a concise conclusion and practical recommendations highlighting the potential role of haemoglobin polymorphism in improving productivity, adaptability, and herd management of crossbred dairy cattle. We also thank the reviewer for the insightful comment regarding the potential association between haemoglobin (Hb) polymorphism and performance traits. We agree that examining such associations is important. In our study, we collected comprehensive phenotypic data, including udder morphology, mastitis incidence, milk production, and physiological and heat stress indicators. However, the relatively small sample size and extremely unbalanced distribution of genotypes limited our ability to perform meaningful statistical association analyses. We have now highlighted in the manuscript that future studies with larger and more balanced populations, coupled with molecular characterization of Hb genes, are needed to validate observed polymorphisms and clarify their implications for genetic improvement. Additionally, we have recommended that breeding programs consider Hb polymorphism as a genetic marker to enhance both productivity and adaptability in crossbred dairy cattle, and that extension programs raise farmer awareness of its potential role.

Reviewer: 2

Reviewers comments Authors response

1. Line 24–25: Misinterpretation of FIS values. FIS= 0.38 and 0.41 are described as “inbreeding coefficient value of the HF crosses relative to the subpopulations grouped into…”, but FIS is not a single value per grouping type—it should be one estimate per subpopulation or an average, not two separate values for “blood levels” and “locations” without clarifying they are separate analyses.

Thank you for pointing out the misinterpretation. We agree that the original wording could be misinterpreted. To clarify, FIS was calculated for each subpopulation however, the reported values are pooled FIS estimates across subpopulations for each grouping scheme. We have revised the manuscript to make this explicit and added a short description of the analysis.

2. Line 26–27: States FST = 0.01 for location and 0.06 for blood level, implying higher genetic differentiation among blood levels—but FST = 0.06 is still very low (near panmixia), yet the phrasing may mislead readers into over-interpreting biological significance.

Thank you. That is a potential over-interpretation of the FST values. Now the revised text states that although the FST for blood level (0.06) is numerically higher than that for location (0.01), both values fall within the range considered very low genetic differentiation.

3. Line 29–30: Concludes a “relatively higher level of similitude” based on low heterozygosity and high homozygosity—this is contradictory: high homozygosity suggests low genetic variation, not necessarily “similitude” across groups. The term is vague and scientifically imprecise.

Thank you for this insightful comment. We agree that the phrase “relatively higher level of similitude” is vague and misinterpreted. To address this, we have revised the text to use more scientifically appropriate terminology.

4. Line 57: Calls hemoglobin an “evergreen red protein”— non-scientific, metaphorical language inappropriate for a research manuscript.

We agree that the phrase “evergreen red protein” is metaphorical and inappropriate for a scientific manuscript. To avoid confusion and maintain scientific accuracy, we have removed the term and replaced it with a precise description of haemoglobin.

5. Line 68–69: States only “a few research works” on Hb polymorphism in Ethiopia, citing Gezahegn (1996) and Pal & Mummed (2014)—but Pal & Mummed (2014) studied Ogaden cattle, not general Ethiopian breeds. This misrepresents scope.

We thank the reviewer for this helpful observation. We agree that our initial wording may have implied that Pal & Mummed (2014) examined general Ethiopian breeds, whereas their study focused specifically on Ogaden cattle. Our intention was to highlight that only limited research has been conducted on haemoglobin polymorphism within Ethiopian cattle populations, and the studies available are breed-specific rather than nationally comprehensive.

6. Line 77–78: Asserts protein polymorphism studies are “easier to implement” than DNA-based methods—but this is context-dependent and outdated (e.g., PCR-based genotyping is now cheaper and more accessible than electrophoresis in many settings).

We value the reviewer's perceptive comment. We concur that the relative simplicity and affordability of DNA-based versus protein-based approaches vary depending on the situation, and that developments in PCR-based genotyping have increased the accessibility and affordability of numerous molecular techniques. However, our goal was to highlight that phenotypic assays like electrophoresis may still be more useful in laboratory settings with limited resources like Ethiopia because they require less equipment and cost less in reagents, rather than to suggest that protein polymorphism techniques are now generally easier or less expensive. We have rephrased the sentence to sound more appropriate.

7. Line 85–86: Hypothesis states Hb frequencies “are under Hardy-Weinberg equilibrium”—but HWE is a null model, not a hypothesis to be confirmed; appropriate phrasing would be testing deviation from HWE.

We thank the reviewer for this important clarification. We agree that Hardy–Weinberg equilibrium (HWE) should not be stated as a hypothesis to be confirmed. We have revised the statement to correctly reflect that the study tested for deviation from the Hardy-Weinberg equilibrium, rather than hypothesizing that the population is in HWE.

8. Line 120–122: Claims 13 cows per blood level per location → 3 blood levels × 3 locations × 13 = 117, which is correct—but later tables report genotype counts that don’t sum correctly (e.g., Table 1: HF87.5% shows 38+1+0 = 39, but 13 per location would be 39 total—OK), yet Table 2 totals per location (e.g., SHA: 35+3+1=39) seem consistent, so this is not an error, but potential confusion arises.

We appreciate the reviewer’s intention to cross-check the sampling structure and numbers presented in Tables 1 and 2. However, the tables do not contradict the sampling design; they simply represent two different categorizations of the same 117 animals—one by blood level (sampling frame), and one by observed haemoglobin genotype (study outcome). To avoid potential confusion, we have added a brief explanatory note in the manuscript.

9. Line 140–143: Uses carbon tetrachloride (CCl4) for hemolysate prep—CCl4 is highly toxic, carcinogenic, and largely obsolete in modern labs. No safety or ethical justification provided despite known hazards.

We appreciate the reviewer bringing up the safety issue. We recognized that organic solvents should be avoided wherever possible and that carbon tetrachloride (CCl₄) is a dangerous solvent. All procedures containing CCl₄ were carried out in compliance with institutional chemical-safety guidelines to reduce risk, including the use of suitable personal protective equipment (masks, gloves, lab coats), safe storage, and disposal of hazardous waste. We have included a clear description of these safety precautions in the Methods. Additionally, we are dedicated to using alternative safe haemolysate preparation techniques in future projects when practical.

10. Line 147–148: Uses agarose gel electrophoresis at pH 8.6 for Hb typing—standard practice uses starch or cellulose acetate gels, not agarose, for Hb isoform separation. Agarose has poor resolution for small charge differences in globins—methodologically questionable.

We agree with the reviewer on cellulose acetate or starch gels being the classical media for haemoglobin electrophoresis due to their high resolving power for slight charge differences. At first, we planned to work on cellulose acetate or starch gels, after facing equipment and chemical inaccessibility we swapped our plan to work on Agarose gel horizontal electrophoresis. We didn’t decide to use agarose gel just because of its availability before checking its universally accepted methodologies that has been successfully applied in both biomedical and veterinary studies (Carlström, G. & Liberg, P., 1975; Giot, J.F., 2010). Furthermore, previous bovine work (Pal & Mummed, 2014) has successfully resolved haemoglobin variants (Hb A, Hb B, and heterozygotes) using alkaline agarose electrophoresis. This supports our choice of agarose at pH 8.6 as providing sufficient resolution for the major bovine Hb polymorphisms.

Carlström, G. & Liberg, P. Agarose Gel Electrophoretic Separation of Blood Serum Proteins in Cattle. Acta Veterinaria Scandinavica, 16:520–524 (1975).

Giot, J.-F. Agarose Gel Electrophoresis — Applications in Clinical Chemistry. Journal of Medical Biochemistry, 29(1):9–14 (2010).

11. Line 155–157: Assigns fast band = HbBB, slow = HbAA—but in cattle, HbB is typically the slow-migrating variant (contrary to human HbS). Literature (e.g., Bachmann et al. 1978) shows HbB (Zebu) migrates slower than HbA (taurine). Likely misassignment of allele identity.

Thank you for pointing this out. We re-examined the literatures and confirmed that our study is In line with previous work on Ogaden cattle agarose electrophoresis revealed three haemoglobin phenotypes (slow, fast and heterozygote). The study reported 54% slow Hb (AA), 12% fast Hb (BB) and 33% moderate Hb AB individuals using agarose gel methods. Biochemical studies (starch gel electrophoresis) also showed that among cattle globins (A, B, etc.), Hb B migrates fastest, than others, with Hb A slower (Efremov, 1965). Consequently, our assignment of the faster band to the B allele and the slower band to the A allele is provisional, based on migration in our agarose system and on how similar Ogaden studies have reported banding patterns. We therefore recommend molecular confirmation (β-globin gene sequencing or targeted PCR/RFLP) to definitively identify alleles in this crossbred population. Meanwhile, Bachmann et al. (1978) reported on the frequency of HbB in Zebu vs taurine cattle, but we couldn’t find specifically reporting slower mobility of HbB in their gel system. Efremov G, Braend M. Differences in cattle globins. Biochem J. 1965 Dec;97(3):867-9.

12. Table 1, HF87.5% row: Expected HbBB = 0, observed = 0—but Chi-square value listed as “1.0ns”—this is mathematically impossible if expected = observed = 0. Likely a software or transcription error.

We thank the reviewer for pointing out the mislabelling in the HF87.5% row of Table 1. The value shown as “1.0ns” in the table was not a χ² statistic but a rounded p-value (p ≈ 1.00) reported by PopGene. PopGene does not compute the χ² statistic using all three genotype classes (AA, AB, BB). Instead, in this subgroup it applied the collapsed χ² test, in which genotypes are grouped into:

• Heterozygotes (AB)

• Homozygotes (AA + BB)

As you know, this approach is common in some population genetics software, especially when the expected number of rare homozygotes is very small (<1), because the standard χ² test becomes invalid under such conditions.

In the HF87.5% subgroup, the expected count for HbBB is extremely low (E=0.00). For this reason, PopGene automatically used the collapsed test and returned only a p-value (≈1.00), which led to the confusion in the original table. After recalculation using genotype-derived allele frequencies, the χ² statistic for this subgroup is effectively 0.00 (df = 1, p=1.00).

Calculation for HF87.5%

• Observed AB = 1

• Expected AB = 1

• Observed homozygotes total (AA+BB) = 38 + 0 = 38

• Expected homozygotes = 38

So:

χ2= (1−1)2/1+(38−38)2/38=0+0=0

• χ² = 1.0 ns (this is actually p = 1.0, not χ² = 1.0)

We have updated Table 1 and 2 to report χ² and p explicitly and added an explanation in the methodology about the exact HWE testing for groups with very small (<1) expected counts.

13. Table 1, Entire population: Observed HbAB = 13, expected = 22.4 → large deviation, Chi-square = 0.00 (p < 0.001)—but 0.00 is not a valid Chi-square value; probably means p = 0.00, but mislabeled.

Thank you, =0.00 was mislabeled and it is p=value not χ². We have updated the table (Table 1) to report χ² and p explicitly.

12. Table 3: Expected heterozygosity (He) for HF50% = 0.32, Observed (Ho) = 0.18 → deficit of heterozygotes, consistent with inbreeding—but no statistical test (e.g., FIS per group) is provided in table, only pooled.

Thank you for this observation. In Table 3, we reported the expected heterozygosity (He) and observed heterozygosity (Ho) for each blood level group, which already allow the calculation of Wright’s inbreeding coefficient (FIS) for each group as FIS = (He− Ho) / He. For example, for the HF50% group (He = 0.32; Ho = 0.18), this corresponds to an FIS of approximately 0.44, indicating a deficit of heterozygotes. Although our table presents only the pooled FIS value, readers can readily compute FIS for each group using these values. Also, including FIS per group would result in a table that’s not manageable.

13. Table 5: Genetic identity >1.0 (e.g., HF50% vs HF50% = 1.0, OK—but HF75% vs HF75% = -0.99? Negative identity? Impossible—values must be 0–1. Typo: likely 0.99, not -0.99.

Thank you. In table 5 and 6 a hyphen is mistaken for negative sign. We used a hyphen for the diagonal used to separate the above and below results. We have updated the tables and replaced with asterisks.

14. Table 6: SHA vs SHA = -1.0—again, identity cannot be negative. Should be 1.0. Formatting error (hyphen misread as minus).

Thank you gain. We have solved the issue

15. Line 263–265: Reports FST= 0.01 (locations) as “significant” and FST= 0.06 (blood levels) as “non-significant”—but no p-values or confidence intervals provided to support “significance” claims.

Thank you for drawing attention to this issue. PopGene reports FST estimates for the defined groups but does not provide accompanying p-values or confidence interva

---

## [Editor Report · Decision Letter 1]

15 Dec 2025

Haemoglobin polymorphism in different blood levels of Ethiopian Zebu x Holstein Friesian crossbred dairy cattle

PONE-D-25-40346R1

Dear Dr. Tesfaye,

We’re pleased to inform you that your manuscript has been judged scientifically suitable for publication and will be formally accepted for publication once it meets all outstanding technical requirements.

Kind regards,

Mourad Mahmoud

Academic Editor

PLOS One
---

## [Editor Report · Acceptance letter]

PONE-D-25-40346R1

PLOS One

Dear Dr. Tesfaye,

I'm pleased to inform you that your manuscript has been deemed suitable for publication in PLOS One. Congratulations! Your manuscript is now being handed over to our production team.

Kind regards,

on behalf of

Dr. Mourad Mahmoud

Academic Editor

PLOS One